# Large-Scale Notification Dispatch with Bundle Treatments and Multi-Outcome Uplift Optimization

Jiajing Xu [1]   Yanyun Li [*1]   Yongbao Song [*1]   Minqin Zhu [1]   Huxiao Ji [1]   Linchuan Li [1]   Cunyi Zhang [1]
Xuanping Li [1]   Kaiqiao Zhan [1]   Yanan Niu [1]

## Abstract

Notification dispatch plays a critical role in large-scale user engagement, involving complex trade-offs across notification timing, presentation style, multiple outcomes, and constraints. In this paper, we formulate it as a constrained optimization over bundle treatments that jointly specify timing and presentation style, aiming to maximize incremental Daily Active Users (DAU) subject to platform-level budget and device vendor-specific quota constraints. The problem is challenging due to multi-dimensional, small-effect uplift estimation and large-scale constrained optimization. To address these challenges, we propose **B**undle **U**plift **O**ptimization with **P**runed **L**agrangian-based **R**elaxation (BUOPLR), a two-stage notification dispatch method that decouples uplift estimation from constrained decision-making. BUOPLR first learns bundle-level, multi-outcome small uplift through an architecture that captures cross-treatment and cross-outcome relationships, and then performs scalable assignment by restricting the decision space and applying Lagrangian relaxation to a small set of global constraints. Offline experiments show BUOPLR outperforms state-of-the-art methods, and online A/B tests increase DAU by 0.5%. BUOPLR is now deployed on Kuaishou, a large-scale short video platform serving over 100 million users daily.

## 1. Introduction

Notification systems are widely adopted by large-scale Internet platforms (Gao et al., 2018; Yancey & Settles, 2020; O'Brien et al., 2022) to optimize notification dispatch for improving user engagement. Prior work shows that notification volume (Tu et al., 2021), timing (Prabhakar et al., 2022), and presentation style (Bahir et al., 2019) jointly influence users' engagement. In practice, mobile notification dispatch is subject to two key constraints: (1) notification quotas imposed by device vendors (e.g., OPPO) to limit dispatch frequency, and (2) monetary budgets under cost-per-click (CPC) pricing, where certain rich notification styles incur charges upon user interaction. Under these constraints, jointly optimizing notification volume, timing, and presentation style is essential for maximizing overall effectiveness, leading to a large-scale resource allocation problem.

Two dominant approaches exist for resource allocation: two-stage pipelines and decision-focused learning. Two-stage methods first estimate individual treatment effects via uplift models, then solve a constrained optimization problem (e.g., knapsack) (Zhao et al., 2019; Jiang et al., 2020; Albert & Goldenberg, 2022; Wang et al., 2023). Decision-focused methods integrate optimization objectives into training, jointly learning outcomes and assignments, e.g., DRP (Du et al., 2019) and DFCL (Zhou et al., 2024). However, existing methods face two challenges in notification dispatch. First, they do not scale to billion-level per-user quotas, as enforcing daily limits induces billions of constraints, making standard Lagrangian relaxation impractical. Second, they fail to model interactions among multi-dimensional bundle treatments and multiple outcomes, leading to biased uplift estimates and suboptimal assignments.

To address these challenges, we propose Bundle Uplift Optimization with Pruned Lagrangian-based Relaxation (BUOPLR), which includes two components: 1) an efficient Lagrangian-based optimization method for large-scale notification dispatch. By constructing a score that jointly reflects marginal engagement gain and cost competitiveness, we rank candidate bundle assignments and aggressively prune the feasible assignment space. This score-based domain reduction ensures that most constraints are satisfied by construction and reduces the number of relaxed constraints to a constant, enabling billion-user–scale optimization to be solved within hours. 2) a novel uplift model that captures structural relationships within the bundle treatment and multi-outcome space. Specifically, the model integrates

*Equal contribution [1]Kuaishou Technology, Beijing, China. Correspondence to: Xuanping Li <lixuanping@kuaishou.com>.

implicit temporal representations, outcome coupling, and monotonicity across presentation styles to hierarchically model multi-dimensional treatment effects. To further address the instability of small uplift estimation under bundle treatments, we introduce a sliding reconstruction mechanism that enables effective information sharing across adjacent treatments, leading to robust and stable uplift modeling.

Our main contributions are summarized as follows:

- **Billion-Scale Constraint Solver:** We address the limitations of traditional resource allocation methods when applied to notification dispatch by developing a solver that efficiently handles *billions* of user-level constraints while remaining computationally tractable.

- **Bundle Treatment and Multi-Outcome Uplift Modeling:** We design an uplift model that jointly captures the effects of composite bundle treatments—combining notification timing and presentation style—across multiple outcomes within a unified architecture.

- **Production Impact:** BUOPLR has been deployed on the PMOS (Push Multi-dimensional Online System) platform at Kuaishou and demonstrates statistically significant improvements. In online A/B testing, it preserves identical business constraints while increasing Daily Active Users (DAU) by 0.5% ($p < 0.01$).

**Related Works.** Existing work related to notification dispatch and resource allocation falls into two main categories, both of which have limitations in large-scale notification systems. Early notification dispatch methods model user response using reinforcement learning or survival analysis (Yuan et al., 2019; Prabhakar et al., 2022; Yuan et al., 2022; O'Brien et al., 2022; Yang et al., 2023), but largely ignore system-level constraints on notification volume. Later systems introduce constraints via staged or heuristic designs, such as enforcing per-user or global quotas without jointly optimizing timing and presentation style (Zhao et al., 2018; Ji et al., 2024), which sacrifices global optimality. More generally, resource allocation approaches have been applied to notification dispatch, often integrating uplift modeling with knapsack-style optimization, typically solved via Lagrangian relaxation (Zhao et al., 2019; Jiang et al., 2020; Albert & Goldenberg, 2022; Wang et al., 2023). However, in notification systems, per-user quota constraints scale to billions of instances, rendering standard Lagrangian methods impractical, while existing uplift models fail to capture interactions across multiple treatment dimensions. In contrast, our approach provides a unified formulation that jointly optimizes multi-dimensional notification decisions under massive per-user and platform-level constraints. A more detailed discussion of related work is provided in Appendix A.

## 2. Problem Formulation

In this paper, we formulate notification dispatch as a constrained optimization problem that simultaneously accounts for device vendor–imposed notification quotas and platform-level notification budget limits. To ensure a positive user experience, a minimum interval between consecutive notifications is enforced by discretizing each user's daily notification window into $H$ time slots. Consider users indexed by $i \in \{1, 2, \ldots, N\}$. Each time slot $h \in \{1, 2, ..., H\}$ can be paired with the $k \in \{1, ..., K\}$ notification presentation style, yielding $HK$ notification bundles indexed by $j \in \{1, \ldots, HK\}$, where each bundle corresponds to a unique pair $(h(j), k(j))$. Let $\mathcal{J}(h) = \{j \mid h(j) = h\}$ and $\mathcal{J}(k) = \{j \mid k(j) = k\}$ denote the sets of bundles associated with time slot $h$ and style $k$, respectively. We define a binary variable $z_{i,j} \in \{0, 1\}$ indicating whether notification bundle $j$ is assigned to user $i$.

**Outcomes and Costs.** Each dispatched notification produces two outcomes: incremental user engagement, measured by incremental DAU, and payment cost upon user click. Let $d_i^{con}$ denote the baseline DAU for user $i$ without notification (i.e., the control condition). For each notification bundle $j$, we define $\Delta d_{i,j}$ as the incremental DAU relative to the baseline, and $\Delta c_{i,j}$ denotes the incremental cost induced by user clicks. DAU is binary, taking values in $\{0, 1\}$, while cost takes values in $\{0, \text{CPC}\}$, where CPC is constant across all users. Without loss of generality, we normalize cost to $\{0, 1\}$.

**Constraints and Objective.** The system is subject to a platform-level notification budget $C$, a global quota $Q$, and per-user, per-style quotas $q_{ik}$ imposed by device vendors. The objective is to maximize total incremental DAU while respecting these constraints:

$$
\begin{aligned}
\max_{z} \quad & \sum_{i=1}^{N} \sum_{j=1}^{HK} z_{i,j} \, \Delta d_{i,j} \\
\text{s.t.} \quad & \sum_{i=1}^{N} \sum_{j=1}^{HK} z_{i,j} \, \Delta c_{i,j} \leq C, \\
& \sum_{i=1}^{N} \sum_{j=1}^{HK} z_{i,j} \leq Q, \quad\quad (1) \\
& \sum_{j \in \mathcal{J}(k)} z_{i,j} \leq q_{i,k}, \; \forall i, \forall k, \\
& \sum_{j \in \mathcal{J}(h)} z_{i,j} \leq 1, \; \forall i, \forall h, \\
& z_{i,j} \in \{0, 1\}, \; \forall i, j,
\end{aligned}
$$

where $\sum_{j \in \mathcal{J}(h)} z_{i,j} \leq 1$ ensures that at most one notification is sent to a user in any given time slot $h$. The constraint $\sum_{j=1}^{HK} z_{i,j} \leq q_{i,k(j)}, \forall i$ enforces vendor-imposed user-level

limits. Each $z_{i,j} = 1$ corresponds to assigning a bundle treatment that jointly specifies notification timing and presentation style; $z_{i,j} = 0$ corresponds to the control condition with no notification.

## 3. Methodology

### 3.1. Constrained Optimization for Notification Dispatch

The optimization problem in Equation (1) is a large-scale knapsack problem and is NP-hard in general. To obtain a tractable solution, prior works (Zhang et al., 2020) adopt a Lagrangian dual–based relaxation to obtain a tractable solution. The resulting Lagrangian function is given by:

$$
\begin{aligned}
\mathcal{L}(z; \lambda, \mu, \gamma) = {} & \sum_{i=1}^{N} \sum_{j=1}^{HK} z_{i,j}\, \Delta d_{i,j} \\
& - \lambda \Big( \sum_{i=1}^{N} \sum_{j=1}^{HK} z_{i,j}\, \Delta c_{i,j} - C \Big) \\
& - \mu \Big( \sum_{i=1}^{N} \sum_{j=1}^{HK} z_{i,j} - Q \Big) \\
& - \sum_{i=1}^{N} \sum_{k=1}^{K} \gamma_{i,k} \Big( \sum_{j \in \mathcal{J}(k)} z_{i,j} - q_{i,k} \Big),
\end{aligned}
\tag{2}
$$

where $z_{i,j} \in \{0,1\}$ and $\sum_{j \in \mathcal{J}(h)} z_{i,j} \leq 1, \forall i, \forall h$. The $\Delta d_{i,j}$ and $\Delta c_{i,j}$ denote the uplift DAU and the corresponding uplift cost, respectively; $\lambda$, $\mu$ and $\gamma_{i,k}$ are Lagrange multipliers for the platform-level budget constraint, device vendor–imposed global notification quota and user-level style-specific quota for user $i$ and presentation style $k$.

The Lagrangian dual function is defined as:

$$
D(\lambda, \mu, \gamma) = \max_{z}\ \mathcal{L}(z; \lambda, \mu, \gamma).
\tag{3}
$$

Collecting all terms in the Lagrangian that depend on $z_{i,j}$, the inner maximization problem can be written as:

$$
\max_{z} \sum_{i=1}^{N} \sum_{j=1}^{HK} z_{i,j} (\Delta d_{i,j} - \lambda\, \Delta c_{i,j} - \mu - \gamma_{i,k(j)}).
\tag{4}
$$

With $z_{i,j} \in \{0,1\}$ and $\sum_{j \in \mathcal{J}(h)} z_{i,j} \leq 1$, this admits a greedy closed-form solution: for each user $i$ and time slot $h$, assign at most one bundle maximizing the adjusted utility:

$$
j^*(i,h) = \arg\max_{j \in \mathcal{J}(h)} (\Delta d_{i,j} - \lambda\, \Delta c_{i,j} - \mu - \gamma_{i,k(j)}).
\tag{5}
$$

The corresponding assignment variables are determined as:

$$
z_{i,j} =
\begin{cases}
1, & \text{if } j = j^*(i,h) \quad \forall i, h, \\
0, & \text{otherwise.}
\end{cases}
\tag{6}
$$

Subsequently, at iteration $t$, the multipliers are updated as:

$$
\begin{aligned}
\lambda^{t+1} &= \lambda^t - \alpha_\lambda\, \nabla_\lambda \mathcal{L}(z^t; \lambda^t, \mu^t, \gamma^t), \\
\mu^{t+1} &= \mu^t - \alpha_\mu\, \nabla_\mu \mathcal{L}(z^t; \lambda^t, \mu^t, \gamma^t), \\
\gamma_{i,k}^{t+1} &= \gamma_{i,k}^t - \alpha_\gamma\, \nabla_{\gamma_{i,k}} \mathcal{L}(z^t; \lambda^t, \mu^t, \gamma^t), \quad \forall i, k,
\end{aligned}
\tag{7}
$$

with step sizes $\alpha_\lambda, \alpha_\mu, \alpha_\gamma$, this iteration progressively refines primal assignments and multipliers, producing near-optimal solutions in practice.

Let $T$ denote the total number of iterations. The time complexity of the Lagrangian dual–based relaxation method is $O(T \cdot N \cdot H \cdot K)$, where each iteration scales linearly with the number of decision variables $z$. For problems at the billion-user scale, this results in a runtime far exceeding one day, rendering the standard Lagrangian dual approach impractical for large-scale notification dispatch.

#### 3.1.1. BILLION-SCALE LAGRANGIAN SOLVER

To reduce the computational cost of relaxing numerous constraints, we enforce user-level, style-specific notification quotas imposed by device vendors directly via a score-based ranking, avoiding additional Lagrange multipliers and keeping the number of dual variables constant, thereby simplifying the dual optimization.

We define a bundle-level score $s_{ij}$ to quantify the priority of assigning bundle $j$ to user $i$:

$$
\begin{aligned}
s_{ij} = {} & \beta(\Delta d_{ij} - \lambda\, \Delta c_{ij}) \\
& + (1 - \beta)[(\Delta d_{ij} - \lambda\, \Delta c_{ij}) \\
& - \max_{\substack{j' \in \mathcal{J}(h(j)) \\ k(j') \neq k(j)}} (\Delta d_{i,j'} - \lambda\, \Delta c_{i,j'})].
\end{aligned}
\tag{8}
$$

The first term measures the net return, while the second captures competitiveness relative to other styles in the same time slot, respecting $\sum_{j \in \mathcal{J}(h)} z_{i,j} \leq 1$. The hyperparameter $\beta \in [0, 1]$ balances net return against competitiveness.

Given the scores $\{s_{ij} \mid j \in \mathcal{J}(k)\}$ of user $i$ for a fixed presentation style $k$ across all $H$ time slots, we define a time-wise ranking function $r(\cdot)$ to compare these scores. Specifically, for each bundle $j \in \mathcal{J}(k)$, we define $r(s_{ij}) \in \{1, 2, \ldots, H\}$ as the rank of $s_{ij}$ when the scores $\{s_{ij} \mid j \in \mathcal{J}(k)\}$ are sorted in descending order:

$$
r(s_{ij}) = \mathrm{rank}(s_{ij}; \{s_{ij} \mid j \in \mathcal{J}(k(j))\}),
\tag{9}
$$

where rank 1 corresponds to the highest score, indicating the relative priority of time slot $h(j)$ is for user $i$ under style $k$ relative to other time slots.

The admissible bundle set is then:

$$
\mathcal{A}_{i,k} = \{\, j \in \mathcal{J}(k) \mid r(s_{ij}) \leq q_{i,k} \,\},
\tag{10}
$$

which retains only the top-$q_{i,k}$ time slots for each style.

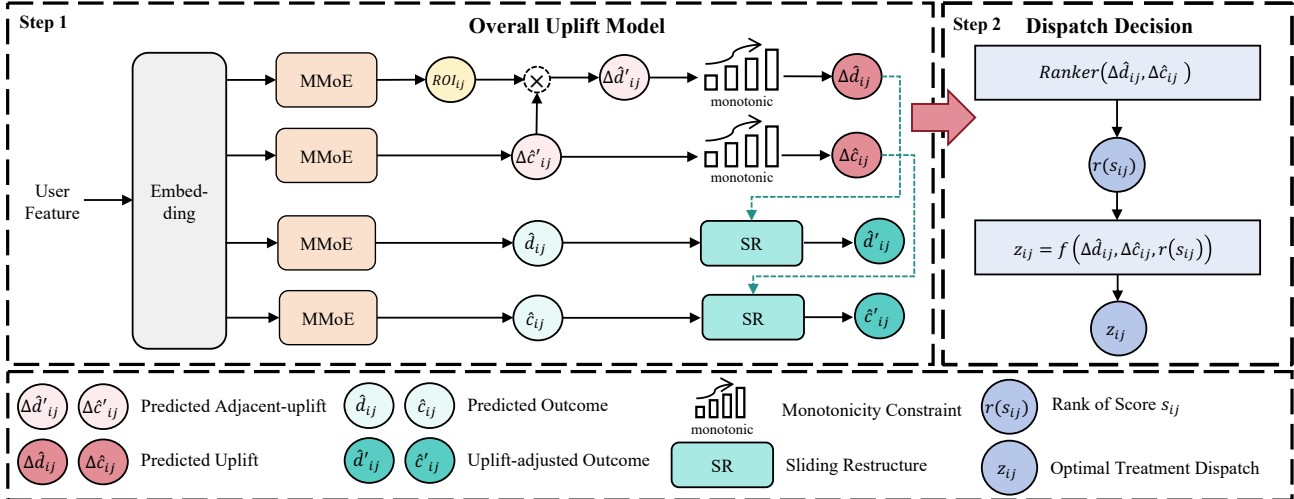

*Figure 1.* Illustration of our Optimal Notification Dispatch. The process has two steps. Step 1: an uplift model estimates multi-outcome uplift for bundled treatments (Time × Style). Using implicit temporal representations, ROI correlations, and monotonicity constraints, the model captures hierarchical interactions across time, outcomes, and styles, producing $\Delta\hat{d}_{ij}$ and $\Delta\hat{c}_{ij}$. Training employs multi-task learning with Sliding-Window Restructuring for cross-treatment outcome reconstruction and direct outcome estimation. Step 2: the Billion-Scale Lagrangian Solver ranks candidates by decision scores $s_{ij}$ to prune low-quality options, solves the relaxed Lagrangian problem, and derives the optimal treatment for each user based on predicted uplift and computed operators.

By construction, this restriction ensures that the user-level quota constraints $\sum_{j\in\mathcal{J}(k)} z_{i,j} \le q_{i,k}$ are always satisfied and therefore need not be included in the Lagrangian relaxation. Let $\mathcal{A} = \bigcup_{i,k} \mathcal{A}_{i,k}$ denote the resulting feasible domain. The optimization problem can be reformulated as:

$$\max_{z} \quad \sum_{i=1}^{N} \sum_{j=1}^{HK} z_{i,j}\, \Delta d_{i,j}$$

$$\text{s.t.} \quad \sum_{i=1}^{N} \sum_{j=1}^{HK} z_{i,j}\, \Delta c_{i,j} \le C,$$

$$\sum_{i=1}^{N} \sum_{j=1}^{HK} z_{i,j} \le Q, \qquad (11)$$

$$\sum_{j\in\mathcal{J}(h)} z_{i,j} \le 1, \quad \forall i, \forall h,$$

$$z_{i,j} = 0, \quad \forall j \notin \mathcal{A},$$

$$z_{i,j} \in \{0,1\}, \quad \forall i, j.$$

Every feasible solution of this reduced problem is also feasible for the original formulation (Equation (1)). Compared to the original problem, this reformulation explicitly prunes the bundle assignment space using the scores $s_{ij}$, retaining only the top $q_{i,k}$ candidates for each user–style pair.

As a result, all user-level vendor quota constraints are satisfied by construction and no longer need to be relaxed. Consequently, the Lagrangian relaxation involves only a constant number of platform-level constraints. Under this reduced formulation, each iteration of the resulting dual

optimization requires a single pass over users and retained bundles, yielding linear time complexity with respect to $N$.

The score-based pruning aligns naturally with the Lagrangian relaxation: $s_{ij}$ captures both the adjusted return of assigning bundle $j$ to user $i$ and its competitiveness against alternative styles at the same time slot, so pruning retains assignments that mimic the greedy maximization of the dual. Approximation errors from domain reduction are confined to bundles near the pruning threshold, which contribute minimally to the objective and are selected only if constraints are nearly tight. In large-scale systems, the fraction of affected assignments is negligible. We therefore treat Equation (11) as an efficient surrogate for the original optimization. A formal discussion of the duality gap is given in Appendix B.

Applying Lagrangian relaxation to the reduced problem yields a dual objective with only two multipliers, $\lambda$ and $\mu$:

$$\mathcal{L}(z; \lambda, \mu) = \sum_{i=1}^{N} \sum_{j=1}^{HK} z_{i,j}\, \Delta d_{i,j}$$

$$- \lambda \Big( \sum_{i=1}^{N} \sum_{j=1}^{HK} z_{i,j}\, \Delta c_{i,j} - C \Big) \qquad (12)$$

$$- \mu \Big( \sum_{i=1}^{N} \sum_{j=1}^{HK} z_{i,j} - Q \Big).$$

For fixed $(\lambda, \mu)$, the dual maximization admits a solution:

$$j^*(i, h) = \arg \max_{j \in \mathcal{J}(h)} [\Delta d_{i,j} - \lambda \Delta c_{i,j} - \mu$$
$$- \mathbb{I}(r(s_{i,j}) > q_{i,k(j)})], \quad (13)$$

with assignments $z_{i,j} = 1$ if $j = j^*(i, h)$ and 0 otherwise.

The multipliers are updated via (sub)gradient steps:

$$\lambda^{t+1} = \lambda^t - \alpha_\lambda \nabla_\lambda \mathcal{L}(z^t; \lambda^t, \mu^t),$$
$$\mu^{t+1} = \mu^t - \alpha_\mu \nabla_\mu \mathcal{L}(z^t; \lambda^t, \mu^t). \quad (14)$$

With $T'$ iterations, the overall complexity is $O(T' \cdot N \cdot K \cdot H \cdot \log H)$, where the $\log H$ term arises from ranking bundles within each $(i, k)$ group. Since only two global constraints are relaxed, the dual is low-dimensional and converges quickly. Empirically, $T'$ typically ranges from 100 to 200 iterations, and does not scale with the problem size. As a result, the overall computational complexity is effectively linear in the number of users $N$. In practice, we also utilize parallel computing and rapid small-data iterations, reducing the solution time to an hourly scale. The pseudocode of the billion-scale Lagrangian solver is provided in Appendix D.

## 3.2. Uplift Model

A key component of our optimization framework is estimating bundle-level uplift terms $\Delta d_{i,j}$ and $\Delta c_{i,j}$, which measure the incremental effects on DAU and cost when bundle $j$—defined by time slot $h(j)$ and style $k(j)$—is dispatched to user $i$. The modeling task is challenging for two reasons. First, the treatment space is multi-dimensional: each treatment corresponds to a bundle that combines notification timing and presentation style, inducing structural relationships across bundle treatments. Second, the model must jointly predict uplift for multiple outcomes, while the true uplift signals are typically small and noisy.

To address these challenges, we design an uplift model that (i) captures structural relationships within the bundle treatment space and across multiple outcomes, and (ii) accurately predicts small uplift effects in large-scale settings.

### 3.2.1. MODELING BUNDLE TREATMENTS–OUTCOMES RELATIONSHIPS

The proposed model estimates bundle-level uplift for DAU and cost across all combinations of notification timing and presentation style. Since the downstream optimization (Equation (1)) depends on these estimates, the model must capture not only uplift magnitudes but also their structural variation across bundle treatments and outcomes.

We focus on three key relationships. **(i) DAU–cost trade-off.** As dispatch assignments are jointly determined by $\Delta d$ and $\Delta c$, we characterize their interaction via incremental return on investment (ROI) for assigning bundle $j$ to user $i$ as $ROI_{i,j} = \Delta d_{i,j} / \Delta c_{i,j}$, and encourage the model to learn their relative relationship by ROI rather than treating them independently. **(ii) Monotonicity across presentation styles.** For a fixed user $i$ and time slot $h$, richer styles generally induce stronger engagement; accordingly, the uplift is assumed to increase monotonically with the style index $k$, and the model is designed to respect this structural prior across styles. **(iii) User-level temporal patterns**. Users exhibit personalized and recurring daily rhythms that affect notification responsiveness. These temporal regularities are latent and user-specific, making explicit parametric modeling impractical; instead, our model learns them implicitly from data to enable personalized, timing-aware uplift estimation.

To capture structural relationships between bundle treatments and outcomes, our uplift model combines implicit representation learning with explicit relational modeling.

**(i) Implicit representation learning.** For each user $i$, we first extract behavioral features from historical interaction data through a shared embedding layer. To capture user-specific temporal response patterns, we adopt a Multi-gate Mixture-of-Experts (MMoE) architecture (Ma et al., 2018), in which individual experts specialize in distinct temporal segments and a gating network aggregates their outputs conditioned on the user embedding. The MMoE yields latent representations across all $H$ time slots. To disentangle temporal effects from outcomes and presentation styles, we instantiate a separate MMoE module for each outcome–style pair. This results in $2K$ modules that jointly span the full (outcome, time, style) space, enabling each module to focus on a well-defined sub-problem while collectively preserving the expressiveness needed to model their interactions.

**(ii) Explicit relational enhancement.** Building on the implicit representations, we introduce explicit relational structures aligned with the downstream dispatch objective. To model the interaction between engagement uplift and cost uplift, the model predicts $ROI_{ij}$ together with $\Delta c_{ij}$, and recovers $\Delta d_{ij}$ via $\Delta d_{ij} = ROI_{ij} \cdot \Delta c_{ij}$. This formulation emphasizes ROI-consistent relative signals rather than independently fitting the magnitudes of $\Delta d$ and $\Delta c$. To enhance style monotonicity, we adopt a two-step strategy. First, we estimate the adjacent uplift and enforce a non-negativity constraint on it. Then, we construct the final uplift via cumulative summation, thereby explicitly satisfying the monotonicity constraints.

### 3.2.2. SMALL UPLIFT ESTIMATION

In large-scale notification systems, uplifts are inherently small and noisy, a challenge further amplified by the bundle treatment space. Standard Two-Model approaches estimate treated and control outcomes separately and compute uplift via differencing, but outcome estimation errors often domi-

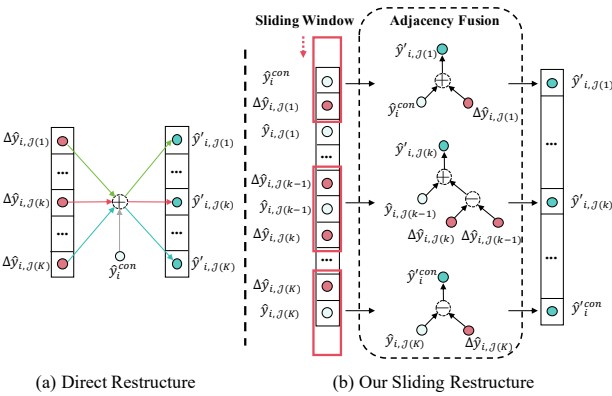

*Figure 2.* Comparison of outcome reconstruction methods under multiple treatments: (a) reconstruction based on the uplift definition using control group outcomes; and (b) reconstruction via a sliding window with a size of 3 and a stride of 2. Applied to the sorted predicted uplift and outcomes, this method reconstructs outcome estimates based on local adjacency relationships within the window.

nate the true uplift signal, leading to unstable predictions.

To address this issue, we design the model to directly learn uplift effects rather than implicitly inferring them through outcome differencing. Since only the outcome corresponding to the actually dispatched bundle is observed for each user, the counterfactual outcomes of other potential bundles remain unobserved even in RCT data. To enable learning under this missing-counterfactual setting, we construct an uplift-adjusted outcome $\hat{y}'_{ij}$ ($y \in \{\text{DAU, Cost}\}$) by reconstructing observable outcomes from predicted uplifts:

$$\hat{y}'_{i,\mathcal{J}(k)} = \Delta\hat{y}_{i,\mathcal{J}(k)} + \hat{y}_i^{con}, \quad \forall i, h, \tag{15}$$

where all bundle treatments share a common control group.

However, as shown in Figure 2 (a), the supervision in this control-centric construction is limited to the dispatched bundle and the control group. It fails to leverage cross-treatment samples. This limitation, similar to that of the Two-Model approach, highlights the key challenge of enabling effective cross-treatment information sharing. To address this limitation, we exploit the monotonic structure along the presentation style dimension and introduce auxiliary supervision by observable outcomes, enabling robust uplift learning despite sparsity and noise.

**Leveraging monotonic structure across styles.** Bundle treatments exhibit an inherent monotonic relationship along the presentation style dimension. This assumption is based on established empirical findings from online experiments on notification systems, with further details provided in Appendix C.1. Exploiting this structure, we reconstruct uplift-adjusted outcomes using adjacent treatments via a sliding window over styles, as illustrated in Figure 2(b). First, we

interleave the predicted uplift and predicted outcome in ascending order of $k$. Next, we traverse this sequence using a sliding window of size 3 with a stride of 2. For each window, we calculate the uplift-adjusted outcome via adjacency fusion as follows:

$$
\hat{y}'_{i,\mathcal{J}(k)}
= \begin{cases}
\hat{y}_i^{con} + \Delta\hat{y}_{i,\mathcal{J}(1)}, & \text{if } k = 1, \\
\hat{y}_{i,\mathcal{J}(k-1)} + \Delta\hat{y}_{i,\mathcal{J}(k)} - \Delta\hat{y}_{i,\mathcal{J}(k-1)}, & \text{otherwise},
\end{cases}
\tag{16}
$$

with $\hat{y}'^{con}_i = \hat{y}_{i,\mathcal{J}(K)} - \Delta\hat{y}_{i,\mathcal{J}(K)}$. The uplift loss reconstructs the observed outcome using mean squared error:

$$\mathcal{L}_{\text{uplift}}(y^{\text{true}}, \hat{y}') = \text{MSE}(y^{\text{true}}, \hat{y}'). \tag{17}$$

This design induces explicit dependencies across adjacent styles and chains all bundle treatments along the style dimension, significantly improving information sharing and noise robustness.

**Auxiliary outcome supervision.** To ensure stable learning of treatment-specific responses, we additionally introduce a direct outcome prediction task. For binary outcomes, its loss is defined as follows:

$$
\begin{aligned}
&\mathcal{L}_{\text{direct}}(y^{\text{true}}, \hat{y}) \\
&= -\frac{1}{N}\sum_{i=1}^{N}[y^{\text{true}}\log\hat{y} + (1 - y^{\text{true}})\log(1 - \hat{y})].
\end{aligned}
\tag{18}
$$

This auxiliary task anchors the model predictions to observable outcomes and mitigates error accumulation from chained uplift estimation.

**Overall training objective.** The model jointly learns uplift and outcome prediction for both DAU and cost. The overall loss is formulated as follows:

$$
\begin{aligned}
\mathcal{L}_{\text{total}}&(d^{\text{true}}, \hat{d}, \hat{d}', c^{\text{true}}, \hat{c}, \hat{c}') \\
&= \mathcal{L}_{\text{direct}}^d(d^{\text{true}}, \hat{d}) + \mathcal{L}_{\text{uplift}}^d(d^{\text{true}}, \hat{d}') \\
&\quad + \mathcal{L}_{\text{direct}}^c(c^{\text{true}}, \hat{c}) + \mathcal{L}_{\text{uplift}}^c(c^{\text{true}}, \hat{c}').
\end{aligned}
\tag{19}
$$

## 4. Experiments

In this section, we conduct extensive experiments to evaluate the effectiveness of the proposed BUOPLR method on a synthetic dataset, the public CRITEO dataset, and a large-scale industrial notification dataset. Our evaluation focuses on three research questions:

- **RQ1:** How does BUOPLR compare with baseline methods in offline uplift estimation and dispatch evaluation?

*Table 1.* Comparison of overall uplift modeling performance between BUOPLR and baseline methods on the synthetic, public, and notification datasets. Best results are highlighted in bold.

| Method | Synthetic Dataset | | CRITEO Dataset | | Notification Dataset | |
|---|---|---|---|---|---|---|
| | AUUC | QINI | AUUC | QINI | AUUC | QINI |
| S-learner | 0.0504 | 0.0594 | 0.0053 | 0.1634 | 0.0054 | 0.0049 |
| Causal Forest | 0.0491 | 0.0561 | 0.0054 | 0.1668 | 0.0057 | 0.0052 |
| DFCL | 0.0584 | 0.0687 | 0.0053 | 0.1625 | 0.0057 | 0.0052 |
| BUOPLR | **0.0888** | **0.1064** | **0.0057** | **0.1753** | **0.0089** | **0.0081** |

- **RQ2:** How do the key design components of BUOPLR affect its performance?

- **RQ3:** How does BUOPLR perform in real-world industrial notification scenarios?

## 4.1. Dataset Description

**Synthetic Dataset.** We construct a synthetic dataset for offline evaluation of notification dispatch. The dataset mimics real notification scenarios with time–style bundle treatments and DAU and cost outcomes. Unlike real-world data, it provides full counterfactual outcomes, enabling precise evaluation of dispatch performance. The data generation process is designed to satisfy three properties: (1) user characteristics are independent of treatment assignment; (2) for any time slot, uplift is monotonic with respect to presentation style; and (3) user responses exhibit explicit temporal dependencies across time slots. Details are provided in Appendix C.2.

**Public Dataset.** We use the CRITEO dataset (Diemert et al., 2018), which contains 13.9 million samples from a randomized controlled trial. The dataset involves a binary treatment with two potential outcomes, visit and conversion. We randomly split the data into 90% training and 10% testing sets. Following (Zhou et al., 2023), visit is treated as cost and conversion as value for resource allocation modeling.

**Notification Dataset.** We collected RCT data from a 1% user sample of a major short-video platform in China, spanning the first two weeks of July 2025. The last day is used for testing, and the preceding days for training. The dataset defines 40 bundle treatments (20 time slots × 2 presentation styles), with 3.5 million samples per treatment. Each user receives one treatment per day, and two outcomes are recorded: DAU and cost. Notification volume, budget, and negative feedback rates are strictly controlled. User behavior features from the previous 30 days are used for prediction. This dataset enables evaluation of both bundle assignment and multi-outcome uplift estimation.

## 4.2. Evaluation Metrics and Methods

**Evaluation Metrics.** For offline evaluation, we assess uplift modeling and dispatch performance using:

- **AUUQC (Area Under the Uplift or Qini Curve):** Measures treatment ranking quality (Gutierrez & Gérardy, 2017; Zhang et al., 2021; Zhu et al.) via the Uplift Curve (AUUC) (Gutierrez & Gérardy, 2017) and Qini Curve (QINI) (Devriendt et al., 2020). Formal definitions are in Appendix C.3.

- **AUCC (Area Under the Cost Curve):** Evaluates ROI-oriented ranking under budget constraints (Ai et al., 2022; Zhou et al., 2023; 2024), reflecting how efficiently cost converts to incremental outcomes.

For online evaluation, we use the following metrics:

- **DAU**: Daily Active Users.

- **CTR**: Click-through rate of dispatched notifications.

- **Watch Time**: Total duration spent on watching videos.

- **NDRR**: Next-Day Retention Rate.

**Baselines and Implementation.** Offline comparisons include Causal Forest (Athey & Wager, 2019) and S-Learner (Künzel et al., 2019) for uplift estimation, and DFCL (Zhou et al., 2024) for dispatch evaluation. To respect user-level constraints, we apply a unified truncation strategy to baseline outputs. Online comparisons are conducted against an industrial baseline, namely TimeWheel, a multi-stage system that has been iteratively optimized on the Kuaishou platform over multiple years. This baseline system tackles resource-constrained notification dispatch via a multi-stage framework, which sequentially determines three core components: (1) notification volume, (2) timing, and (3) style. For the timing modeling, we follow (Ji et al., 2024) by predicting user activity across time slots and dispatching notifications at users' most probable active moments. For quota allocation and presentation style optimization, we follow the framework proposed in (Tu et al., 2021). Specifically, causal forests are utilized to estimate incremental DAU and

cost, followed by clustering and linear programming for final resource allocation.

The implementation details of our uplift model are provided in Appendix C.4. For the training phase, we use a batch size of 1024 and optimize the model with Adam, using an initial learning rate of $1 \times 10^{-5}$, $\beta_1 = 0.9$, $\beta_2 = 0.999$, and $\epsilon = 10^{-8}$. The Hyperparameter studies on the learning rate are in Appendix C.5.

### 4.3. Offline Experiment (RQ1)

The overall uplift modeling performance evaluation is presented in Table 1. On the CRITEO dataset, our model outperforms all baselines with a $7\% - 8\%$ relative gain. For the more challenging Notification and Synthetic datasets, performance drops for all models, yet BUOPLR achieves the best results, improving $60$–$80\%$ over baselines.

*Table 2.* AUCC evaluation results on the CRITEO dataset. The evaluation of DFCL is taken from (Zhou et al., 2024).

| Method | AUCC | Improvement |
|---|---|---|
| S-learner | 0.6490 | / |
| Causal Forest | 0.7440 | $+14.6\%$ |
| DFCL | 0.7859 | $+21.1\%$ |
| **BUOPLR** | **0.8030** | **$+23.7\%$** |

For offline dispatch evaluation, BUOPLR demonstrates superior performance across both synthetic and CRITEO datasets. On the synthetic dataset (Figure 3), it consistently achieves the highest Delta DAU for any Delta Cost, showing strong dispatch efficiency. While most models perform adequately under low-budget regimes by targeting top-uplift users, BUOPLR excels at ranking medium-to-low uplift users, delivering substantial gains under higher budgets. On the CRITEO dataset (Table 2), BUOPLR outperforms traditional uplift models and resource allocation method DFCL, effectively capturing the intrinsic value–cost relationship.

### 4.4. Ablation Study (RQ2)

We conduct ablation studies to evaluate the impact of key components in our uplift model:

- **w/o ROI**: Removes the ROI prediction network, using DAU's uplift directly, to assess the performance of modeling outcome relationships.

- **w/o MTT**: Disables **M**ulti-**T**ask **T**raining and retains only the uplift prediction. The $\mathcal{L}_{direct}$ loss is removed to evaluate multi-task learning impact.

- **w/o MON**: For each treatment $j$, we predict uplift directly relative to the control, bypassing the monotonicity module based on adjacent uplifts, to evaluate the importance of monotonic structure across styles.

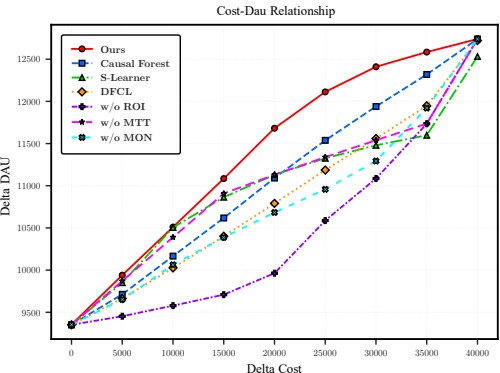

*Figure 3.* Comparison of the delta DAU values under delta cost constraints in resource allocation experiments on synthetic datasets.

*Table 3.* Ablation studies for BUOPLR on the synthetic dataset.

| Method | AUUC | QINI | AUCC |
|---|---|---|---|
| **BUOPLR** | **0.0888** | **0.1064** | **0.6479** |
| w/o ROI | 0.0281 | 0.0323 | 0.4853 |
| w/o MTT | 0.0491 | 0.0582 | 0.4984 |
| w/o MON | 0.0535 | 0.0638 | 0.5517 |

As shown in Table 3 and Figure 3, removing any component substantially reduces uplift ranking and dispatch performance. The ROI network links DAU and cost, multi-task learning stabilizes estimation across small uplifts, and monotonicity constraints enforce expected ordering across styles. Together, these elements markedly enhance performance in bundle treatment and multi-outcome scenarios.

In addition to the synthetic data constructed based on bundle treatment and multi-outcome scenario, we also conducted ablation studies on public Criteo datasets and reached consistent conclusions. The detailed results are provided in Appendix C.6.

*Table 4.* The experimental results from Online A/B tests.

| Method | DAU | CTR | Watch Time | NDRR |
|---|---|---|---|---|
| TimeWheel | / | / | / | / |
| BUOPLR | +0.504% | +3.936% | +0.095% | +0.22% |

### 4.5. Online A/B test (RQ3)

As demonstrated in Table 4, BUOPLR outperforms the baseline method TimeWheel, which is described in Section 4.2, on all evaluation metrics. Specifically, on the platform side, DAU increases by $0.5\%$. On the user side, CTR rises by $3.9\%$, indicating higher notification receptivity. Furthermore, gains in Watch Time ($+0.095\%$) and Next-Day Retention Rate (NDRR, $+0.22\%$) underscore enhanced user

engagement and platform affinity. Experiments validate that the proposed ranking-based pruning solver and bundle uplift model improve notification dispatch performance, bringing mutual benefits to users and the platform.

## 5. Conclusion

In this paper, we propose BUOPLR, a novel method for large-scale notification dispatch. We formalized the problem as maximizing incremental DAU via bundle treatments under global and user-level constraints. BUOPLR is a two-stage method: first, a Lagrangian-based solver that efficiently computes notification assignments within a restricted decision space. We propose a Lagrangian dual sorting-based pruning technique to address the prohibitive computational cost caused by billion-scale constraints; and second, we propose an uplift model designed for bundle treatment and multi-outcome scenarios. By leveraging multi-layer relation learning and cross-treatment sample training, it effectively addresses the challenges of complex structural relationships in multi-dimensional spaces and small uplift signals, achieving stable and robust uplift estimation. Experimental results show that BUOPLR consistently outperforms existing methods in both offline and online tests, achieving a $0.5\%$ DAU gain, and has been deployed on the PMOS platform at Kuaishou to serve over 100 million users.

## Limitations

In this work, we assume that notification effects are temporally independent to focus on large-scale assignment optimization and bundle treatment uplift estimation. While global resource constraints partially enable allocation over time, explicitly modeling sequential dependencies and temporal dynamics remains an important direction for future work. Extending the framework to support more diverse treatment types and real-time streaming decisions, as well as incorporating users' historical notification sequences, are key next steps for improving notification dispatch methods.

## Impact Statement

This work contributes to notification system and online marketing by addressing two long-standing challenges in optimal resource allocation and uplift modeling: scalable optimization under billion-scale constraints and reliable uplift estimation in complex settings with bundle treatments and multiple outcomes. Our paper makes the following key contributions: 1) To address large-scale optimization, we develop a Lagrangian dual ranking-based pruning technique that significantly cuts down slack variables, facilitating the solution of billion-scale constrained knapsack problems in a matter of hours; 2) For uplift modeling, we present a new architecture designed for bundle treatments and multi-outcome scenarios. This model captures implicit and explicit structural relationships in multi-dimensional space via MMoE-based temporal representation learning, outcome cross-features, and monotonicity constraints. Furthermore, it incorporates a sliding restructure strategy to maximize cross-treatment sample utility and bolster direct uplift learning, effectively mitigating the small uplift issue inherent in granular bundle treatment outcomes.

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

## A. Detailed Related Work

### A.1. Notification Dispatch Optimization

Prior research in notification dispatch optimization has evolved through three primary paradigms, each with distinct limitations. Early notification system employed reinforcement learning (Prabhakar et al., 2022; Yuan et al., 2022; O'Brien et al., 2022) or survival analysis (Yuan et al., 2019; Yang et al., 2023) to model the notification dispatch, which ignore systemic constraints on the individual and overall volume of notifications, making them unsuitable for the context of this study. Subsequent notification system introduce constraint through divergent strategies. Pinterest's system (Zhao et al., 2018) implements global notification volume constraints combined with per-user notification quota allocation, but crucially omits optimization of notification timing. TIM (Ji et al., 2024) considers notification timing optimization through a two-stage approach, first allocating user-level notification quotas and then optimizing notification timing. However, this two-stage approach sacrifices global optimality by treating notification quota assignment and dispatch as independent processes. Meanwhile, studies on global constraint optimization (Gupta et al., 2017) address platform-level notification budgets but fail to model spatiotemporal interactions between notifications. In contrast to these approaches, our method formulates notification optimization as a unified decision-making problem that jointly accounts for user-level quota constraints, platform-level capacity limits, and multi-dimensional notification decisions. Rather than decoupling quota allocation from dispatch notifications, our method performs joint optimization over notification assignment and timing under all relevant constraints, thereby preserving global optimality.

### A.2. Optimal Resource Allocation

Two branches of methods have been widely used for resource allocation: two-stage pipelines and decision-focused learning. Two-stage approaches first estimate individual treatment effects (ITEs) using uplift models to quantify the causal impact of treatments, then solve a constrained optimization problem (e.g., knapsack) to assign treatments under resource limits (Jiang et al., 2020; Albert & Goldenberg, 2022; Zhao et al., 2019; Wang et al., 2023). Decision-focused methods integrate optimization objectives into model training, jointly learning outcomes and treatment assignments; for example, DRP (Du et al., 2019) directly predicts ROI for ranking-based assignments, while DFCL (Zhou et al., 2024) uses Lagrange multipliers to achieve constrained optimality guarantees.

Despite their successes, existing methods face two key challenges in notification dispatch. First, they do not scale to per-user quota constraints at billion-user scale, as enforcing daily limits generates billions of constraints and Lagrangian relaxation becomes computationally impractical. Second, they fail to capture interactions among multi-dimensional bundle treatments (time × presentation style) and multiple outcomes, leading to biased uplift estimates and suboptimal treatment assignments.

The second challenge lies in the limitations of existing uplift modeling techniques used in the first stage. These models can be broadly categorized into tree-based methods and neural network–based methods. Representative tree-based approaches include Uplift Trees (Rzepakowski & Jaroszewicz, 2012) and Causal Forests (Athey & Wager, 2019), which partition individuals into subpopulations that exhibit heterogeneous responses to a given treatment. Neural network–based approaches, such as the S-learner and T-learner (Künzel et al., 2019), typically adopt a two-model strategy that estimates potential outcomes for treatment and control groups separately and computes uplift as their difference.

To address these challenges, our method adopts a unified optimization framework that simultaneously scales to billion-level user constraints and supports multi-dimensional treatment modeling. Specifically, we design a scalable Lagrangian solver capable of operating under per-user constraints instantiated at massive scale, together with an uplift modeling architecture that explicitly models interaction effects across multiple treatment dimensions. This unified formulation enables efficient approximation of near-optimal, constraint-compliant notification decisions in large-scale deployment scenarios.

## B. Analysis of the Approximate Solution

This appendix quantifies the optimality loss introduced by the score-based domain reduction in Section 3.1.1. We show that, although pruning replaces the original problem (Equation (1)) by the smaller problem (Equation (11)), the *relative* optimality gap vanishes as the user population grows, so the pruned problem is an asymptotically tight surrogate at notification scale.

**Notation.** We work with the partial Lagrangian relaxation in which only the platform-level budget $C$ is dualized; all remaining constraints are kept in primal form. For a multiplier $\lambda \geq 0$, define

$$\mathcal{L}_o(\lambda) = \max_{z \in \mathcal{Z}_o} \sum_{i,j} z_{i,j}(\Delta d_{i,j} - \lambda \Delta c_{i,j}) + \lambda C,$$

$$\mathcal{L}_n(\lambda) = \max_{z \in \mathcal{Z}_n} \sum_{i,j} z_{i,j}(\Delta d_{i,j} - \lambda \Delta c_{i,j}) + \lambda C,$$

where $\mathcal{Z}_o$ and $\mathcal{Z}_n$ are the feasible sets of the original problem Equation (1) and the pruned problem Equation (11), respectively (i.e. $\mathcal{Z}_n$ adds the domain restriction $z_{i,j} = 0$ for $j \notin \mathcal{A}$). Let

- $p_o^*, p_n^*$ denote the optimal primal values of $\mathcal{L}_o(\lambda)$ and $\mathcal{L}_n(\lambda)$;

- $L_o^* = \mathcal{L}_o(\lambda_o^*)$ and $L_n^* = \mathcal{L}_n(\lambda_n^*)$ denote the optimal dual values, where $\lambda_o^*$, $\lambda_n^*$ are the corresponding optimal multipliers.

**Theorem B.1** (Sandwich relation between primal and dual values). *For the partial Lagrangian relaxation $\mathcal{L}_o(\lambda)$ and $\mathcal{L}_n(\lambda)$,*

$$p_n^* \leq p_o^* \leq L_o^*.$$

*Proof.* We prove the two inequalities separately.

**(i) $p_n^* \leq p_o^*$.** Pruning only restricts the feasible set, i.e. $\mathcal{Z}_n \subseteq \mathcal{Z}_o$. Hence every feasible solution of $\mathcal{L}_n(\lambda)$ is feasible $\mathcal{L}_o(\lambda)$, and the optimal value of the more constrained problem cannot exceed that of the less constrained one.

**(ii) $p_o^* \leq L_o^*$.** Standard weak duality gives $p_o^* \leq L_o^*$. Combining (i) and (ii) yields Theorem B.1. $\square$

**Asymptotic vanishing of the relative gap.** Theorem B.1 alone does not bound the discrepancy between $p_n^*$ and $p_o^*$. We now show that, as the user population $N$ grows, the *relative* gap $(p_o^* - p_n^*)/L_o^*$ vanishes. Decomposing this ratio yields

$$0 \leq \underbrace{\frac{p_o^* - p_n^*}{L_o^*}}_{\text{relative gap}} \leq \underbrace{\frac{L_o^* - L_n^*}{L_o^*}}_{\text{(I) pruning gap}} + \underbrace{\frac{L_n^* - p_n^*}{L_o^*}}_{\text{(II) duality gap}},$$

where the first inequality follows from Theorem B.1 and the equality from $L_o^* - p_n^* = (L_o^* - L_n^*) + (L_n^* - p_n^*)$. We bound the two terms separately.

**(I) Pruning gap.** Let $\mathcal{E}_o, \mathcal{E}_n$ be the optimal solution sets of $\mathcal{L}_o(\lambda_o^*)$ and $\mathcal{L}_n(\lambda_n^*)$, and denote by $\mathcal{J}_{\text{out}} = \mathcal{E}_o \setminus \mathcal{E}_n$ the set of *user–bundle assignments that are optimal for the original Lagrangian but excluded by pruning*. For each excluded assignment, replacing it with the best feasible alternative inside $\mathcal{Z}_n$ incurs a value loss bounded by a constant $\bar{e}$, where $\bar{e}$ depends only on the per-bundle uplift range $\max_{i,j}|\Delta d_{i,j}| + \lambda_o^* \max_{i,j}|\Delta c_{i,j}|$ and is therefore independent of $N$. Hence

$$L_o^* - L_n^* \leq |\mathcal{J}_{\text{out}}| \cdot \bar{e}.$$

It remains to bound $|\mathcal{J}_{\text{out}}|$. We illustrate the argument for $K = 1$.

- *Unconstrained regime.* When the global resource is slack, the original-problem optimum is the per-user greedy top-$q_i$ selection, where $q_i$ is the per-user quota. Because the bundle score $s_{ij}$ in Equation (8) is monotonically aligned with the dual-adjusted utility $\Delta d_{i,j} - \lambda \Delta c_{i,j}$, the score-based pruning recovers the same set, so $\mathcal{J}_{\text{out}} = \emptyset$.

- *Constrained regime.* When the global budget is binding, the Shapley–Folkman theorem (Starr, 1969) states that the non-convexity loss of a separable mixed-integer program with $m$ coupling constraints is of order $O(m)$, provided the per-user feasible sets are uniformly bounded—a condition met here since $z_{i,j} \in \{0, 1\}$ and $|\mathcal{J}| = HK$ is finite. In our formulation, the relaxation involves only a constant number $m$ of platform-level constraints (budget and global quota), independent of $N$. Consequently $|\mathcal{J}_{\text{out}}| = O(m) = O(1)$.

Combining these two regimes yields $L_o^* - L_n^* = O(1)$. Since $L_o^*$ is a sum of per-user incremental DAU contributions, $L_o^* = \Theta(N)$, and therefore

$$\frac{L_o^* - L_n^*}{L_o^*} = \frac{O(1)}{\Theta(N)} \xrightarrow{N \to \infty} 0.$$

**(II) Duality gap.** The pruned problem is again a separable knapsack-type program with a constant number of coupling constraints. Applying the Shapley–Folkman theorem to its Lagrangian relaxation gives an absolute duality gap of $O(1)$ (Starr, 1969). Together with $L_o^* = \Theta(N)$ (and $L_n^* \leq L_o^*$, so the denominator is the right normalizer), this yields

$$\frac{L_n^* - p_n^*}{L_o^*} = \frac{O(1)}{\Theta(N)} \xrightarrow[N \to \infty]{} 0.$$

**Conclusion.** Combining the pruning-gap and duality-gap bounds derived above gives

$$0 \leq \frac{p_o^* - p_n^*}{L_o^*} \leq \frac{O(1)}{\Theta(N)} + \frac{O(1)}{\Theta(N)} \xrightarrow[N \to \infty]{} 0.$$

Hence the relative optimality loss induced by score-based pruning vanishes at rate $O(1/N)$. At notification scale ($N$ on the order of $10^9$), the bound above certifies that the pruned problem is an essentially tight surrogate of the original problem, while admitting a billion-scale-friendly solver.

# C. Experimental Details and Additional Results

This appendix provides the technical details that support the empirical study in Section 4. We (i) state and empirically verify the monotonicity assumption used by our uplift model (Section C.1), (ii) describe the synthetic data-generating process used for the counterfactual and dispatch evaluation (Section C.2), (iii) recall the formal definitions of the AUUC and Qini curves used as offline metrics (Section C.3), (iv) document the implementation of our uplift model (Section C.4), and (v) report additional hyper-parameter and ablation studies (Section C.5, Section C.6).

## C.1. Monotonicity Assumption Across Presentation Styles

**Assumption (Style monotonicity).** Let $\tau_{i,k}^{(h)} = \mathbb{E}[Y_i \,|\, \mathrm{do}(h, k)] - \mathbb{E}[Y_i \,|\, \mathrm{do}(\emptyset)]$ denote the conditional average treatment effect (CATE) of dispatching style $k$ at time slot $h$ to user $i$, where $Y$ is the outcome of interest (DAU or cost). For any fixed time slot $h \in \{1, \ldots, H\}$ and any user sub-population $\mathcal{S}$,

$$\tau_{\mathcal{S},k}^{(h)} \leq \tau_{\mathcal{S},k+1}^{(h)}, \qquad k = 1, \ldots, K - 1,$$

i.e. richer presentation styles induce non-decreasing treatment effects on engagement.

**Empirical verification.** We validate this assumption on the RCT data underlying our notification dataset. The RCT population is randomly partitioned into 20 disjoint subgroups in order to expose any local violations that would be averaged out at the population level. For every subgroup we observe that (i) the average treatment effect of each style is non-decreasing in $k$; (ii) the incremental uplift between adjacent styles, i.e. $\tau_{\mathcal{S},k+1}^{(h)} - \tau_{\mathcal{S},k}^{(h)}$, lies in the $[50\%, 150\%]$ range of $\tau_{\mathcal{S},k}^{(h)}$; and (iii) the notification disable rate is statistically indistinguishable across styles. Together, these observations empirically substantiate the style-monotonicity assumption and justify its use as a structural prior in the uplift architecture (Section 3.2.2).

## C.2. Synthetic Dataset Generation

The synthetic dataset is designed to satisfy the three properties listed in Section 4.1: (i) covariates are independent of treatment assignment; (ii) for any time slot, uplift is monotone in the presentation style; and (iii) outcomes exhibit non-trivial temporal dependence across slots. A sample is a tuple $(\mathbf{x}_i, j_i, d_i, c_i)$, where $\mathbf{x}_i \in \mathbb{R}^p$ is the user feature vector, $j_i$ is the bundle assignment, and $(d_i, c_i)$ are the realized DAU and cost outcomes.

### C.2.1. USER FEATURE GENERATION

Each user is represented by a $p$-dimensional continuous covariate vector $\mathbf{x}_i = (x_{i,1}, \ldots, x_{i,p})^\top$ with $x_{i,\ell} \overset{\text{i.i.d.}}{\sim} \mathcal{N}(0,1)$. We use $p = 32$.

### C.2.2. TREATMENT GENERATION

We instantiate the bundle space with $H = 2$ time slots and $K = 2$ presentation styles (weak and strong), giving $HK = 4$ bundles indexed by $j \in \{1, 2, 3, 4\}$, each corresponding to a unique pair $(h(j), k(j))$. To simulate randomized assignment

and eliminate confounding, the indicator that bundle $j$ is dispatched is sampled independently as $T_{i,j} \sim \text{Bernoulli}(0.5)$. The realized treatment vector $\mathbf{T}_i \in \{0, 1\}^{HK}$ satisfies the time-slot exclusivity constraint $\sum_{j \in \mathcal{J}(h)} T_{i,j} \leq 1$ by post-hoc projection.

### C.2.3. DAU RESPONSE GENERATION

To enforce monotonicity across styles within each time slot and to capture the dependence between weak and strong treatments, we partition each user into one of three latent response strata, following standard principal-stratification terminology:

- **Persuadables** ($\text{per}_j$): users that benefit *only if* bundle $j$ is dispatched.

- **Sure-things** ($\text{sur}_j$): users that benefit *regardless* of dispatch.

- **Lost-causes** ($\text{los}_j$): users that do *not* benefit regardless of dispatch.

The latent stratum probabilities for bundle $j$ are generated from the covariates by

$$\text{per}_j(\mathbf{x}) = \theta_j \, \cos\left(\sum_{\ell=1}^p x_\ell\right) + \delta_j,$$

$$\text{sur}_j(\mathbf{x}) = \theta_j \, \sin\left(\sum_{\ell=1}^p x_\ell\right) + \delta_j,$$

$$\text{los}_j(\mathbf{x}) = \theta_j \, \sum_{\ell=1}^p (\cos(x_\ell) + \eta_j),$$

where $\theta_j, \delta_j, \eta_j$ are bundle-specific scalars that control the relative magnitudes of the three strata. For each $(h, k)$ we choose $(\theta_j, \delta_j, \eta_j)$ such that the strong style strictly dominates the weak style in every stratum at the same time slot, which embeds the monotonicity assumption (Section C.1) into the data-generating process. The realized binary DAU outcome $d_{i,j}$ is then sampled from the stratum-induced response distribution.

### C.2.4. COST RESPONSE GENERATION

Cost is incurred only under intervention; we therefore restrict the cost mechanism to dispatched bundles. Following the same template as Section C.2.3 but omitting the sure-things term (since baseline users incur no cost), we generate

$$\text{perc}_j(\mathbf{x}) = \theta_j \, \cos\left(\sum_{\ell=1}^p x_\ell\right) + \delta_j,$$

$$\text{losc}_j(\mathbf{x}) = \theta_j \, \sum_{\ell=1}^p (\cos(x_\ell) + \eta_j).$$

The cost magnitude per dispatched bundle scales with the treatment intensity $k(j)$, ensuring that stronger styles are also more expensive, consistent with the production setting.

### C.3. Definitions of Uplift and Qini Curves

For completeness we recall the two ranking metrics used throughout the offline evaluation. Let $\mathcal{D}$ denote an RCT test set and let $R^T(\mathcal{D}, k)$, $R^C(\mathcal{D}, k)$ be the cumulative outcome of the top-$k$ users in the model-induced ranking, restricted to the treatment and control groups respectively; $N^T(\mathcal{D}, k)$, $N^C(\mathcal{D}, k)$ are the corresponding group sizes. The two curves are defined in Table 5. The Qini curve corrects for treatment/control size imbalance and is therefore preferred when the propensity is non-uniform; the AUUC has a direct interpretation as cumulative incremental outcome and is preferred for ranking tasks at fixed sampling rates.

### C.4. Implementation Details of the Uplift Model

**Input features.** We use 700 statistical features derived from each user's 30-day behavioural history. Categorical features are mapped to 5-dimensional embeddings; the embeddings of all features are concatenated, layer-normalized, and projected through a fully-connected layer with ReLU activation to a 128-dimensional shared user representation.

*Table 5.* Uplift and Qini curves used to evaluate ranking quality.

| Curve Name | Equation of Curve |
|:---:|:---:|
| AUUC | $(\frac{R^T(D,k)}{N^T(D,k)} - \frac{R^C(D,k)}{N^C(D,k)}) \cdot (N^T(D,k) + N^C(D,k))$ |
| QINI | $R^T(D,k) - R^C(D,k)\frac{N^T(D,k)}{N^C(D,k)}$ |

**Network architecture.** The shared user representation is fed into the MMoE module described in Section 3.2.1. The module instantiates 14 experts, each a two-layer MLP with hidden sizes $[128, 64]$ and ReLU activations. For every task, a softmax gating network combines the expert outputs into a task-specific representation, which is then passed to a two-layer MLP head with hidden sizes $[64, 32]$ and a sigmoid output for the binary prediction.

**Training configuration.** Training uses mini-batch size 1024 with the Adam optimizer ($\beta_1 = 0.9$, $\beta_2 = 0.999$, $\epsilon = 10^{-8}$) and an initial learning rate of $1 \times 10^{-5}$, selected by the sweep reported in Section C.5. Loss weights for the uplift and direct-outcome heads are 1.0 for both DAU and cost.

### C.5. Hyper-parameter Study: Learning Rate

We sweep the learning rate over $\{10^{-1}, 10^{-3}, 10^{-5}, 10^{-7}\}$ on the synthetic dataset, keeping all other settings fixed. Table 6 reports AUUC, QINI and AUCC. Both excessively large and excessively small step sizes degrade all three metrics: large step sizes prevent the model from resolving the small uplift signal, while small step sizes lead to under-fitting within the training budget. Performance is maximized at $1 \times 10^{-5}$, which is therefore adopted in all reported experiments.

*Table 6.* Hyper-parameter study on the learning rate of the uplift model on the synthetic dataset.

| Learning Rate | AUUC | QINI | AUCC |
|:---:|:---:|:---:|:---:|
| 1e-1 | 0.0690 | 0.0807 | 0.5650 |
| 1e-3 | 0.0785 | 0.0913 | 0.5832 |
| 1e-5 | **0.0888** | **0.1064** | **0.6479** |
| 1e-7 | 0.0726 | 0.0855 | 0.5751 |

### C.6. Ablation Studies on the CRITEO Dataset

The CRITEO benchmark contains a single binary treatment, so the style-monotonicity component (w/o MON) is not applicable. We therefore restrict the ablation to the two components that are well-defined in the binary setting: the ROI prediction head (w/o ROI) and the multi-task training objective (w/o MTT). Table 7 shows that removing either component causes a clear drop in both AUUC and QINI, mirroring the trends observed on the synthetic dataset (Table 3 of the main paper). This confirms that the benefits of ROI-coupled prediction and auxiliary outcome supervision are not artefacts of the bundle setting: they generalize to standard binary-treatment uplift benchmarks as well.

*Table 7.* Ablation studies for BUOPLR on the Criteo dataset.

| Method | AUUC | QINI |
|:---:|:---:|:---:|
| BUOPLR | **0.0057** | **0.1753** |
| w/o ROI | 0.0051 | 0.1558 |
| w/o MTT | 0.0050 | 0.1520 |

## D. Pseudocode of Billion-Scale Lagrangian Solver

---

**Algorithm 1** Billion-Scale Lagrangian Solver of BUOPLR

---

**Input:**
- Users $\{1, 2, ..., N\}$
- Bundle set $J = \{1, 2, ..., H \times K\}$
- Predicted uplift values $\Delta \hat{d}_{i,j}, \Delta \hat{c}_{i,j}$ for all $i, j$
- User-style quotas $q_{i,k}$
- Platform budget $C$
- Global notification quota $Q$
- Trade-off coefficient $\beta \in [0, 1]$
- Initial multipliers $\lambda^0 \geq 0, \mu^0 \geq 0$
- Step sizes $\alpha_\lambda, \alpha_\mu$
- Number of dual iterations $T'$

**Output:**
- Dispatch decision $z_{i,j} \in \{0, 1\}$

1: Initialize $\lambda \leftarrow \lambda^0, \mu \leftarrow \mu^0$
2: **for** $t = 1$ to $T'$ **do**
3:    // Step 1: score-based domain reduction
4:    **for** user $i = 1$ to $N$ **do**
5:      **for** bundle $j \in J$ **do**
6:        Compute adjusted return: $g_{i,j} \leftarrow \Delta \hat{d}_{i,j} - \lambda \Delta \hat{c}_{i,j}$
7:      **end for**
8:      **for** bundle $j \in J$ **do**
9:        Compute competitiveness term: $comp_{i,j} \leftarrow g_{i,j} - max_{j' \in J(h(j)), k(j') \neq k(j)} g_{i,j'}$
10:       Compute score: $s_{i,j} \leftarrow \beta g_{i,j} + (1 - \beta) comp_{i,j}$
11:      **end for**
12:      **for** style $k = 1$ to $K$ **do**
13:        Rank bundles $j \in J(k)$ by $s_{i,j}$ in descending order and keep the top $q_{i,k}$ bundles
14:        $A_{i,k} \leftarrow$ top $q_{i,k}$ bundles in $J(k)$ by $s_{i,j}$
15:      **end for**
16:    **end for**
17:    Construct the reduced feasible set: $A \leftarrow U_{i=1}^N U_{k=1}^K A_{i,k}$
18:    // Step 2: dual optimization on the reduced problem
19:    Initialize $z_{i,j} \leftarrow 0$ for all $i, j$
20:    **for** user $i = 1$ to $N$ **do**
21:      **for** time slot $h = 1$ to $H$ **do**
22:        $j^*(i, h) \leftarrow argmax_{j \in A_i J(h)}(g_{i,j} - \mu)$
23:        Set $z_{i,j^*(i,h)} \leftarrow 1$ if $j^*(i, h)$ is feasible
24:      **end for**
25:    **end for**
26:    // Step 3: compute global constraint residuals and subgradient update
27:    $cost_{residual} \leftarrow C - \sum_i \sum_j z_{i,j} \Delta \hat{c}_{i,j}$
28:    $quota_{residual} \leftarrow Q - \sum_i \sum_j z_{i,j}$
29:    $\lambda \leftarrow \max(0, \lambda - \alpha_\lambda cost_{residual})$
30:    $\mu \leftarrow \max(0, \mu - \alpha_\mu quota_{residual})$
31: **end for**
32: **return** $z$

---

