# OpenReview forum: "Large-Scale Notification Dispatch with Bundle Treatments and Multi-Outcome Uplift Optimization"
_ICML.cc/2026/Conference — ICML 2026 regular_

### Official Review · Reviewer_Nvmu · 2026-03-11

**Soundness:** 3
**Presentation:** 2
**Significance:** 3
**Originality:** 3
**Overall Recommendation:** 4
**Confidence:** 4

**Summary:**

The paper introduces BUOPLR (Bundle uplift optimization with pruned Lagrangian-based relaxation), a two-stage framework to optimize notification dispatch for platforms. The authors motivate that traditional systems struggle to balance user engagement with complex constraints, especially for "bundle treatments" in which we need to combine both notification timing and style. To address this, they  develop a uplift model that captures implicit structure better. The work's main contribution is a billion-scale constraint solver that makes large-scale resource allocation computationally tractable. They implement a score-based ranking and pruning technique and reduce the optimization task to a constant number of global constraints that can be solved via Lagrangian relaxation. Experimental on synthetic and real-world datasets (Criteo) show improvements over benchmarks and online A/B tests show a 0.5% increase in Daily Active Users (DAU) and a 3.9% rise in Click-Through Rate (CTR).

**Compliance With Llm Reviewing Policy:**

Affirmed.

**Key Questions For Authors:**

1) What baseline is used in Table 4 and what were the numbers for the baseline?
2) Are the numbers in Table 4 for the Criteo dataset statistically significant?
3) Can you explain the "leveraging monotonic structure" in more detail, especially L314 onward?

**Limitations:**

Yes

**Strengths And Weaknesses:**

Strengths:
- BUOPLR addresses large scale constrains by using score-based pruning keeping the number of dual variables constant and computationally tractable.
- Compared to related works the have seperate treatment modeling, their model uses structural priors such as correlations between DAU and cost and monotonicity across presentation styles.
- Their solver is deployed on a platform with 100M users and they have statistically sig increase on DAU, CTR.

Weaknesses:
- The algorithm is not presented in a single location but is spread across the methodology and appendix sections. The optimization is difficult to parse, and the authors have not provided pseudocode or the solver's source code in the supplementary materials, so their reproducibility and generalization to other notification systems seems limited.
- They assume that notification effects are temporally independent. This is a limitation for long-term user behavior dynamics, do related works address this or is it a limitation for them too?

---

> ### Author Rebuttal · Authors · 2026-03-31
>
> Thank you for your positive feedback. We will address your concerns and questions one by one.
>
> **R1 to W1:** \
> We apologize for any confusion caused by our presentation. The complete pseudocode of our method is given below and will be added to the revised version of the paper.
>
> Algorithm 1 Pruned Lagrangian Relaxation of BUOPLR\
> Input:
>  - Users $1,2,...,N$
>  - Bundle set $J=\\{1, 2, ..., H×K\\}$
>  - Predicted uplift values $\Delta \hat d_{i,j},\Delta \hat c_{i,j}$ for all $i, j$
>  - User-style quotas $q_{i,k}$
>  - Platform budget $C$
>  - Global notification quota $Q$
>  - Trade-off coefficient $β\in[0,1]$
>  - Initial multipliers $λ^0≥0,μ^0≥0$
>  - Step sizes $α_λ,α_μ$
>  - Number of dual iterations $T'$
>
> Output:
>  - Dispatch decision $z_{i,j}∈\\{0,1\\}$
>
> Procedure:
>
> 1: Initialize $λ←λ^0,μ←μ^0$\
> 2: for $t=1$ to $T'$ do\
> 3: // Step 1: score-based domain reduction\
> 4: for user $i=1$ to $N$ do\
> 5: for bundle $j∈J$ do\
> 6: Compute adjusted return:\
> 7: $g_{i,j}←\Delta \hat d_{i,j}-λ\Delta \hat c_{i,j}$\
> 8: end for\
> 9: for bundle $j∈J$ do\
> 10: Compute competitiveness term:\
> 11: $comp_{i,j}←g_{i,j}-max_{j'∈J(h(j)),k(j')\neq k(j)}g_{i,j'}$\
> 12: Compute score:\
> 13: $s_{i,j}←β·g_{i,j}+(1-β)·comp_{i,j}$\
> 14: end for\
> 15: for style $k=1$ to $K$ do\
> 16: Rank bundles $j∈J(k)$ by $s_{i,j}$ in descending order and keep the top $q_{i,k}$ bundles\
> 17: $A_{i,k}←$ top $q_{i,k}$ bundles in $J(k)$ by $s_{i,j}$\
> 18: end for\
> 19: end for\
> 20: Construct the reduced feasible set:\
> 21: $A←U_{i=1}^N U_{k=1}^K A_{i,k}$\
> 22: // Step 2: dual optimization on the reduced problem\
> 23: Initialize $z_{i,j}←0$ for all $i, j$\
> 24: for user $i=1$ to $N$ do\
> 25: for time slot $h=1$ to $H$ do\
> 26: $j^* (i,h)←argmax_{j∈A_i ∩ J(h)} (g_{i,j}-μ)$\
> 27: Set $z_{i,j^* (i,h)}←1$ if $j^* (i,h)$ is feasible\
> 28: end for\
> 29: end for\
> 30: // Step 3: compute global constraint residuals and subgradient update\
> 31: $cost_{residual}←C-\sum_i \sum_j z_{i,j} \Delta \hat c_{i,j}$\
> 32: $quota_{residual}←Q-\sum_i \sum_j z_{i,j}$\
> 33: $λ←max(0,λ-α_λ·cost_{residual})$\
> 34: $μ←max(0,μ-α_μ·quota_{residual})$\
> 35: end for\
> 36: return $z$
>
> Furthermore, as mentioned in our Response 2 to Reviewer BGMW, we have also provided the detailed model hyperparameter settings. We hope that these additional details will help improve the reproducibility of our work.
>
> **R2 to W2:** \
> Due to the character limit, please refer to our Response 1 to Reviewer ap32 or Response 3 to Reviewer BGMW, where we discuss similar concerns.
>
> **R3 to Q1:** \
> The baseline in our online experiments tackles resource-constrained notification dispatch via a multi-stage framework: it sequentially determines (1) notification quota, (2) timing, and (3) style. Timing follows Tim[1] by predicting user activity over time slots and sending notifications at the most likely active time. Quota and style are optimized using uplift modeling, where a causal forest estimates incremental DAU and cost, followed by clustering and linear programming for allocation.
>
> Compared to our method, this baseline decomposes a complex problem into a sequence of simpler, hierarchically connected subproblems, at the cost of global optimality and with potential error accumulation.
>
> In online experiments, effectiveness is evaluated by relative improvement over the baseline; thus, absolute baseline values are omitted. Our method achieves a 0.5% DAU lift (p < 0.1), indicating statistical significance.
>
> ### Reference
>
> [1] Ji, H., etc. Tim: Temporal interaction model in notification system. In Proceedings of the 2024 International Conference on Multimedia Retrieval, 2024.
>
> **R4 to Q2:** \
> On the Criteo dataset, although our method slightly outperforms the baseline, the differences are not statistically significant. We attribute this to the relative simplicity of the Criteo dataset, which limits the observable gains across different methods.
>
> **R5 to Q3:** \
> In uplift modeling, each treatment is supported by at most a very limited number of samples, since each sample only observes the outcome under a single treatment. This issue is exacerbated in notification scenarios due to weak uplift signals, making sample efficiency critical.
>
> To address this, we introduce structured dependencies among treatments, allowing each sample to influence multiple treatment heads. The key challenge is modeling relationships among bundle treatments.
>
> From RCT data, we observe a monotonic pattern of uplift with respect to style, which provides a strong prior. Based on this, we propose a sliding reconstruction mechanism (Figure 2) to establish dependencies between adjacent treatment heads along this monotonic dimension.
>
> Intuitively, under monotonicity, estimates at one point can anchor neighboring ones, enabling knowledge sharing across treatments and improving sample efficiency.
>
> We appreciate the feedback and are happy to provide further clarification. If our response addresses your concerns, we would be grateful for reconsideration of the evaluation scores.

---

> > ### Author Rebuttal · Reviewer_Nvmu · 2026-04-07
> >
> > Thank you for the clarifications, however, my original evaluation still stands.

---

> > > ### Author Response · Authors · 2026-04-08
> > >
> > > Thank you very much for your careful review and for your positive evaluation of our work! We sincerely appreciate your time and recognition.

---

### Official Review · Reviewer_BGMW · 2026-03-12

**Soundness:** 4
**Presentation:** 3
**Significance:** 4
**Originality:** 4
**Overall Recommendation:** 5
**Confidence:** 4

**Summary:**

This paper studies the problem of notification dispatch.
The authors point out that due to the constraints of notification quotas and monetary budgets, current notification dispatch methods suffer from a large-scale resource allocation problem.
Traditional Lagrangian dual-based relaxation methods suffer from high time complexity and are hard to apply in large-scale notification dispatch scenarios.
In order to address such a problem, this paper proposes Bundle Uplift Optimization with Pruned Lagrangian-based Relaxation (BUOPLR). Different from the traditional Lagrangian dual-based relaxation method, BUOPLR is a more efficient method that is designed for large-scale dispatch scenarios. The core idea of BUOPLR is to first estimate the bundle-level uplift scores and then simplify the constrained optimization problem by pruning the bundle assignment space using a score-based Lagrangian relaxation strategy.
This paper conducts experiments on three kinds of datasets, including both synthetic and real-world datasets, to evaluate the effectiveness of the proposed method. The results show that the proposed method outperforms the traditional methods on both dispatch performance and efficiency.

**Compliance With Llm Reviewing Policy:**

Affirmed.

**Key Questions For Authors:**

N/A

**Limitations:**

yes

**Strengths And Weaknesses:**

## Strengths

1. This paper solved a problem in the notification dispatch, the inefficiency in the large-scale setting, which is important in real-world applications.
2. This paper formulates the problem as a clear constraint optimization problem, and also presents the idea of the proposed method under the Lagrangian optimization framework, which makes the presentation easy to understand.
3. This paper uses strong theory to prove the claims, including the limitations of traditional methods, the idea of proposed methods, and the theoretical performance. These theories make the claims in this paper strong.
4. This paper conducts experiments in different kinds of datasets, including a synthetic dataset, the public dataset CRITEO, and a large-scale industrial notification dataset. These comprehensive experiments further show the effectiveness of the proposed method.
5. This paper deploys the proposed method BUOPLR on a major internet platform and shows improvements, which show great production impact in real-world applications.



## Weaknesses

1. Hyperparameter study: While this paper has reported the experimental results, it has not discussed the influence of hyperparameters such as the learning rate of the optimizer.
2. Implementation details: This paper doesn't give enough implementation details in experiments, especially about the uplift model. As the paper says the proposed method has been applied in a real-world system, more implementation details can help with reproducibility.
3. Overstrong assumptions: This paper assumes the notification effects are temporally independent, while this constraint is too hard in the real-world application, as the previous notification may influence the attitude of users toward the next notification. Therefore, the dependency between notifications can also be an important factor to consider.

---

> ### Author Rebuttal · Authors · 2026-03-31
>
> Thank you for your positive feedback. We will address your concerns and questions one by one.
>
> **R1 to W1:** \
> We have added a hyperparameter study on the learning rate of the uplift model, and the results are as follows.
> |Learning Rate|AUUC|QINI|AUCC|
> |:---|:---|:---|:---|
> |1e-1|0.0690|0.0807|0.5650|
> |1e-3|0.0785|0.0913|0.5832|
> |1e-5|0.0888|0.1064|0.6479|
> |1e-7|0.0726|0.0855|0.5761|
> |||||
>
> The experimental results indicate that performance declines when the learning rate is set either too low or too high, with the best performance achieved at 1e-5. We will include the above results in the appendix of the final version of the paper.
>
> **R2 to W2:** \
> For the input features and network architecture, we use 700 statistical features derived from users’ behaviors over the past 30 days. In the embedding stage, each feature is first projected into a 5-dimensional embedding vector. The resulting embeddings are then concatenated, followed by layer normalization and a fully connected layer that outputs a 128-dimensional representation with ReLU activation.
>
> For the MMOE module, the expert head consists of hidden layers with sizes [128,64], while each task-specific head has hidden layer sizes [64,32]. The number of experts is set to 14. A sigmoid activation function is applied at the output layer.
>
> We will include the above results in the appendix of the final version of the paper.
>
> **R3 to W3:** \
> Thank you for your concern. Our adoption of the temporal independence assumption is motivated by the following considerations. Notification dispatching, as a sequential large-scale resource allocation problem, is inherently NP-hard. To obtain a practical approximate solution, we adopt a Lagrangian dual formulation, which provides a convex relaxation of the original problem and enables us to derive a meaningful bound. However, applying Lagrangian duality requires independence across decisions. Therefore, following Tim[1], we adopt the temporal independence assumption, under which the allocation of notification quotas over time can be relaxed to a convex optimization problem. Building on this formulation, we focus on two key problems: large-scale assignment optimization and bundle-treatment uplift estimation.
>
> Moreover, we also believe that relaxing the temporal independence assumption and preserving temporal dependencies is an important direction for improving decision quality.
>
> In future work, we will explore how to incorporate temporal effects into large-scale resource allocation problems.
>
> ### Reference
>
> [1] Ji, H., Yang, H., Li, L., Zhang, S., Zhang, C., Li, X., and Ou, W. Tim: Temporal interaction model in notification system. In Proceedings of the 2024 International Conference on Multimedia Retrieval, pp. 1120–1124, 2024.
>
> Thank you once again for your valuable feedback. If you have any further concerns or questions, we are always happy to address them. If you feel that our responses have addressed your concerns, we would appreciate it if you could consider raising your evaluation scores.

---

> > ### Author Rebuttal · Reviewer_BGMW · 2026-04-02
> >
> > Thank you for the authors' response. I think my concerns have been addressed. I believe it's appropriate to maintain my score.

---

> > > ### Author Response · Authors · 2026-04-08
> > >
> > > Thank you for your detailed review, positive feedback, and recommendation. We appreciate your time and glad to have addressed all your questions.

---

### Official Review · Reviewer_YzSE · 2026-03-12

**Soundness:** 2
**Presentation:** 3
**Significance:** 2
**Originality:** 2
**Overall Recommendation:** 4
**Confidence:** 3

**Summary:**

A Bundle Treatments and Multi-Outcome Uplift Optimization.

**Compliance With Llm Reviewing Policy:**

Affirmed.

**Final Justification:**

I maintain my score.

**Key Questions For Authors:**

see above

**Limitations:**

see above

**Strengths And Weaknesses:**

Contributions and Strengths This paper investigates the optimization problem of bundle treatment in Notification Dispatch. The proposed **Bundle Uplift Optimization** avoids the constraint enumeration problem in large-scale data through a novel two-stage solution strategy and feasible region pruning. It also addresses the estimation challenge of uplift with low signal-to-noise ratio by directly modeling uplift rather than outcome differences. Theoretically, the authors demonstrate approximate equivalence between the pruned approximation problem and the original problem. Experimentally, the paper balances offline evaluation with online A/B testing, with method effectiveness validated through real-world deployment—a highly influential and impressive achievement. Weaknesses / Questions 1. In Section 3.2.1, the method explicitly introduces a monotonicity assumption regarding presentation style. This assumption may not hold in certain practical scenarios, particularly for users in poor states or with high historical negative feedback. The authors could consider potential biases arising when this assumption fails, or supplement with empirical validation, such as the proportion of cases where monotonicity holds in real-world data.     2. The objective function formulates total uplift as the sum of uplifts from all assigned bundle treatments. The underlying assumption of additivity (for multiple notification effects) may be stronger than the defined temporal independence. Yet diminishing marginal effects—or even substitution—are common. For instance, if the first notification successfully re-engages an inactive user, the incremental value of a second notification may diminish to zero. I encourage the authors to discuss this alongside temporal independence, or at least acknowledge it in the limitations section. 3. The ablation study in Section 4.4 presents results only on synthetic datasets. As noted earlier, the synthetic data generation process itself explicitly encodes structural priors consistent with certain assumptions. If feasible, supplementing with ablation studies on real-world datasets may provide grounds for generalizability. 4. Theorem B.2 directly uses “≈” as a conclusion format. Even if this aligns with the original statement's logic of “sufficiently close to the optimal value,” it remains a form not favored for rigor. Additionally, the scale of the $O(1)$-order error is unclear, and there is a lack of comparison with similar work.

---

> ### Author Rebuttal · Authors · 2026-03-31
>
> Thank you for your positive feedback. We will address your concerns and questions one by one.
>
> **R1 to W/Q1:** \
> We have two motivations for introducing the monotonicity assumption regarding presentation style.
>
> First, this assumption is supported by our analysis of RCT data from a real-world notification scenario (Lines 357–368 in the manuscript). Specifically, we randomly partitioned the RCT population into 20 groups to examine whether any negative effects would emerge in smaller subpopulations. We found that, within each subgroup, the ATE of each presentation style increases with k, the uplift between adjacent styles increases by 50% to 150%, and the notification disable rate remains unchanged.
>
> Second, monotonicity across treatments is also commonly observed in real-world uplift modeling applications, such as marketing incentives and subsidy allocation. This suggests that our method can be readily transferred to these scenarios.
>
> To alleviate such concerns, we will incorporate the above discussion into the final version of the paper.
>
> **R2 to W/Q2:** \
> Due to the character limit, please refer to our Response 1 to Reviewer ap32 and Response 3 to Reviewer BGMW, where we discuss similar concerns.
>
> The scenario you mentioned, where user activity causes abrupt uplift changes, is indeed a challenging setting for our method. In practice, to mitigate this issue, we simply do not send notifications to users who have already been active on that day. In future work, we will explore how to incorporate temporal effects into large-scale resource allocation problems to fully address this issue.
>
> **R3 to W/Q3:** \
> Ablation results on the public Criteo dataset are presented below.
> |Method|AUUC|QINI|
> |:---|:---|:---|
> |BUOPLR|0.0057|0.1753|
> |w/o ROI|0.0051|0.1558|
> |w/o MTT|0.0050|0.1520|
> |||
>
> The experimental results are consistent with those on the synthetic data, indicating that both the ROI and MTT designs effectively improve the accuracy of uplift estimation. We will include the above results in the appendix of the final version of the paper.
>
> **R4 to W/Q4:** \
> Theorem B.2 aims to show that the original optimization problem and the pruned problem are asymptotically close in terms of their optimal objective values. Thus, the pruning strategy can substantially accelerate computation while preserving performance. To clarify omitted details, we restate key definitions:
> |Symbol|Description|
> |:---|:---|
> |$L_o^*$|optimal value of the original problem|
> |$L_n^*$|optimal value of the pruned problem|
> |$E_o$|optimal solution set of the original problem|
> |$E_n$|optimal solution set of the pruned problem|
> |$J_{out}$=$E_o$-$E_n$|set of original optimal solutions excluded by pruning|
> |$\bar{e}$|the average optimal-value loss contributed by each solution in $J_{out}$|
> ||
>
> Our argument is as follows. In our scenario, the total number of decision variables $N$ is extremely large (billions). If the relative gap between the optimal values of the original problem and the pruned problem, i.e., $(L_o^* -L_n^* )/L_o^*$, vanishes as $N\to\infty$, the effect of pruning is negligible.
>
> The absolute gap $L_o^* -L_n^* $ arises from pruning some original optimal solutions, replaced by suboptimal alternatives. Hence, $L_o^* -L_n^* $ satisfies $O(|J_{out}|\ast\bar{e})$, with $\bar{e}$ independent of $N$.
>
> Assuming independent user-level decisions, we analyze $|J_{out}|$ under $K=1$: 1) When the total resource is unconstrained, the optimal solution to the original problem is simply to greedily select the top-q items for each user, where q denotes the per-user quota. Because the scoring function used in Equation (8) is consistent with the objective of the optimization problem, the pruning step is effectively equivalent to greedy selection. Therefore, the optimization problems before and after pruning yield the same solution. 2) When the total resource is constrained, the Shapley–Folkman theorem implies that the error induced by m global constraints is on the order of O(m). In our scenario, the number of global constraints is a constant. Hence $|J_{out}|$ is also of constant order and does not depend on $N$. Consequently, the absolute gap remains constant as $N$ grows, i.e., it is $O(1)$.
>
> For $L_o^* $, the objective function of optinal problem $L_o$ is defined as the sum of users’ DAU revenues, the value of $L_o^* $ grows with the number of decision variables $N$, and is therefore on the order of $O(N)$. Finally, the relative gap $(L_o^* -L_n^* )/L_o^*=O(1)/O(N)→0$ when $N→+\infty$
>
> Thank you once again for your valuable feedback. If you have any further concerns or questions, we are always happy to address them. If you feel that our responses have addressed your concerns, we would appreciate it if you could consider raising your evaluation scores.

---

> > ### Author Rebuttal · Reviewer_YzSE · 2026-03-31
> >
> > I have no more questions. Thanks for the feedback and I maintain the score.

---

> > > ### Author Response · Authors · 2026-04-08
> > >
> > > We sincerely thank you for the positive evaluation and recommendation. We are delighted that our responses have successfully addressed all your concerns.

---

### Official Review · Reviewer_ap32 · 2026-03-12

**Soundness:** 3
**Presentation:** 3
**Significance:** 3
**Originality:** 3
**Overall Recommendation:** 4
**Confidence:** 3

**Summary:**

This paper investigates the optimization problem of mobile push notification delivery for large-scale internet platforms. In practice, platforms must determine when and how to send notifications to users while simultaneously satisfying multiple objectives and constraints. On the one hand, platforms aim to increase user engagement, such as improving the number of Daily Active Users (DAU). On the other hand, they must operate within predefined click budgets and comply with restrictions imposed by mobile operating systems and device manufacturers regarding notification frequency and user disturbance. These requirements together form a large-scale and highly constrained optimization problem.
To address this challenge, the paper proposes a system framework named BUOPLR. The system decomposes the complex notification allocation problem into two key stages, enabling efficient and scalable optimization.
First, in the prediction stage, the system employs high-precision machine learning models to estimate the potential benefits and associated costs of sending notifications with different formats at different times. These predictions provide a quantitative foundation for subsequent decision-making.
Second, in the optimization stage, the system introduces an efficient scheduling algorithm that rapidly evaluates candidate strategies and prunes suboptimal solutions. As a result, optimization tasks that previously required several days of computation can now be completed within a few hours, significantly improving the overall efficiency of the system.

**Compliance With Llm Reviewing Policy:**

Affirmed.

**Key Questions For Authors:**

Please see the weaknesses.

**Limitations:**

Yes.

**Strengths And Weaknesses:**

1. Soundness
Strength: Clear problem decomposition and solid logical structure
The paper demonstrates a well-structured logical framework. It formulates a complex industrial problem, optimizing notification delivery while avoiding excessive user disturbance, as a large-scale optimization problem with multiple resource and operational constraints. A key design choice is the explicit separation between prediction (estimating benefits and costs) and decision-making (notification allocation), which simplifies the overall problem structure.
To address the computational challenges introduced by billions of constraints, the paper proposes a “score-and-prune followed by centralized optimization” strategy. By filtering low-quality candidate strategies at an early stage, the method substantially reduces the computational burden of the subsequent optimization step. From a theoretical perspective, the authors further analyze the approximation error introduced by the pruning strategy using the Shapley–Folkman lemma in the appendix, arguing that the resulting error becomes negligible at large problem scales. This analysis provides theoretical support for the proposed approximation approach.
Weakness: Simplifying assumption on temporal independence
To ensure computational efficiency in production environments, the model assumes that the effects of notifications at different time points are statistically independent. However, in real-world user behavior, notifications often exhibit temporal dependencies. For instance, receiving a notification earlier in the day may influence a user’s responsiveness to subsequent notifications. Ignoring such sequential effects simplifies the optimization problem but may limit the model’s ability to fully capture realistic user behavior dynamics.

2. Presentation
Strength: Clear organization and effective visualizations
The paper is generally well organized, presenting the problem motivation, methodological design, and experimental evaluation in a coherent progression. Figures and visualizations effectively support the analysis of system performance. For example, in the ablation experiments on synthetic datasets, the line plot illustrating the relationship between investment cost and user activity improvement (Figure 3) clearly demonstrates the relative performance of different approaches. The full model is highlighted with a distinct curve, allowing readers to quickly identify its performance advantage under comparable cost constraints.
Weakness: Limited intuitive explanations for theoretical derivations
While the system design and architecture are described clearly, some of the theoretical analyses in the appendix, such as the derivation of approximation error bounds, are presented primarily through mathematical formulas. Providing additional intuitive explanations or illustrative examples could help readers better understand the practical implications of these theoretical results and improve overall readability.

3. Significance
Strength: Strong practical impact in real-world deployment
A notable strength of this work lies in its practical relevance. Unlike many approaches that are validated only on offline datasets or small-scale experiments, the proposed system has been deployed in a real-world platform serving over 100 million daily active users.
According to the reported online A/B testing results, the system achieved approximately a 0.5% increase in DAU while satisfying all operational constraints. At such a large user scale, this improvement represents substantial business value. Moreover, the simultaneous improvement in click-through rate and user retention suggests that the method enhances platform engagement without causing excessive notification fatigue.

4. Originality
Strength: Effective integration of existing techniques for practical innovation
The originality of the paper mainly lies in its effective integration and adaptation of existing techniques to address practical industrial challenges, rather than proposing entirely new theoretical foundations. By combining several established components in a task-specific manner, the authors manage to overcome key bottlenecks in real-world deployment.
For instance, the predictive model incorporates the MMoE architecture to capture heterogeneous user behavior patterns across different time periods. In addition, the paper employs a sliding reconstruction strategy to address sparsity issues in uplift estimation by leveraging the ordering relationships among notification styles.
Such targeted adaptations of existing techniques to address specific industrial pain points demonstrate a meaningful form of applied innovation and contribute to the practical effectiveness of the proposed system.

---

> ### Author Rebuttal · Authors · 2026-03-31
>
> Thank you for your positive feedback. We will address your concerns and questions one by one.
>
> **R1 to W1:** \
> Thank you for your concern. Our adoption of the temporal independence assumption is motivated by the following considerations. Notification dispatching, as a sequential large-scale resource allocation problem, is inherently NP-hard. To obtain a practical approximate solution, we adopt a Lagrangian dual formulation, which provides a convex relaxation of the original problem and enables us to derive a meaningful bound. However, applying Lagrangian duality requires independence across decisions. Therefore, following Tim[1], we adopt the temporal independence assumption, under which the allocation of notification quotas over time can be relaxed to a convex optimization problem. Building on this formulation, we focus on two key problems: large-scale assignment optimization and bundle-treatment uplift estimation.
>
> Moreover, we also believe that relaxing the temporal independence assumption and preserving temporal dependencies is an important direction for improving decision quality.
>
> In future work, we will explore how to incorporate temporal effects into large-scale resource allocation problems.
>
> ### Reference
>
> [1] Ji, H., Yang, H., Li, L., Zhang, S., Zhang, C., Li, X., and Ou, W. Tim: Temporal interaction model in notification system. In Proceedings of the 2024 International Conference on Multimedia Retrieval, pp. 1120–1124, 2024.
>
> **R2 to W2:** \
> Our intuitive explanation is as follows. The original optimization problem is subject to both user-level constraints and global constraints. When the global constraints are far from active, the optimization should primarily satisfy the user-level constraints. Under this view, the solution can be regarded as retaining, for each user, the decision with the highest gain.
>
> Our pruning strategy proceeds in the opposite direction: rather than directly selecting the highest-utility decisions, we iteratively eliminate the lowest-utility ones. These two dual processes eventually meet at the hyperplane induced by the global constraints. Therefore, the error introduced by pruning is confined to the vicinity of the global-constraint boundary. When the user population N is sufficiently large, this boundary represents only a constant-scale region relative to the entire solution space, suggesting that the approximation error caused by pruning is negligible.
>
> Thank you once again for your valuable feedback. If you have any further concerns or questions, we are always happy to address them. If you feel that our responses have addressed your concerns, we would appreciate it if you could consider raising your evaluation scores.

---

> > ### Author Rebuttal · Reviewer_ap32 · 2026-04-03
> >
> > I will maintain my score.

---

> > > ### Author Response · Authors · 2026-04-08
> > >
> > > We are glad that we have been able to address all of your concerns. We sincerely thank you again for your careful review and for your positive evaluation of our work.

---

### Decision · Program_Chairs · 2026-04-30

**Decision:**

Accept (regular)

**Comment:**

This paper studies the practically relevant problem of notification dispatch in large-scale internet platforms. The reviewers appreciated the real-world impact this work will likely bring. Most of the initial concerns of the reviewers have been resolved after the author rebuttal, which includes clarifications on the assumptions on notification effects, experimental protocol used in the paper, sensitivity to hyperparameter configurations, and ablations on the Criteo dataset.

The main weakness of the paper is that the notification effects are temporally independent, which may not hold in real-world settings. The authors have argued that they need this assumption in order to obtain a practical, approximate solution, which is reasonable. Although this weakness remains, the paper investigates a practically important problem and provides a novel solution. Another weakness (in evaluation) is that the results on real-world data Criteo, are not statistically significant. The authors argued that this is mainly a problem of the dataset rather than the method they proposed, and their argument is reasonable.

Overall, this paper is likely to have a moderate-to-high impact within the research community.